# REWARD MODEL GENERALIZATION FOR COMPUTE-AWARE TEST-TIME REASONING

## ABSTRACT

External test-time reasoning enhances large language models (LLMs) by decoupling generation and selection. At inference time, the model generates multiple reasoning paths, and an auxiliary process reward model (PRM) is used to score and select the best one. A central challenge in this setting is test-time compute optimality (TCO), i.e., how to maximize answer accuracy under a fixed inference budget. In this work, we establish a theoretical framework to analyze how the generalization error of the PRM affects compute efficiency and reasoning performance. Leveraging PAC-Bayes theory, we derive generalization bounds and show that a lower generalization error of PRM leads to fewer samples required to find correct answers. Motivated by this analysis, we propose Compute-Aware Tree Search (CATS), an actor-critic framework that dynamically controls search behavior. The actor outputs sampling hyperparameters based on reward distributions and sparsity statistics, while the critic estimates their utility to guide budget allocation. Experiments on the MATH and AIME benchmarks with various LLMs and PRMs demonstrate that CATS consistently outperforms other external TTS methods, validating our theoretical predictions.

## 1 INTRODUCTION

In recent years, chain-of-thought (CoT) prompting has substantially improved the performance of large language models (LLMs) on complex reasoning tasks such as math problem solving, question answering, and multi-hop retrieval (Wei et al., 2022; Kojima et al., 2022; Yao et al., 2023). To support more effective CoT reasoning, recent works have explored test-time scaling (TTS) strategies that allocate more compute during inference (OpenAI, 2024; DeepSeek-AI, 2025; Team et al., 2025; Snell et al., 2024; Liu et al., 2025; Wu et al., 2024). These approaches can be broadly divided into internal and external ones, with external methods attracting increasing attention due to their ability to enhance performance without modifying the base model (Snell et al., 2024; Liu et al., 2025).

The external TTS framework typically consists of three components: a frozen policy model, a scaling algorithm, and a process reward model (PRM). It generates multiple candidate reasoning paths using the policy model guided by the scaling algorithm (e.g., Best-of-N or Beam Search), and reranks these paths using the PRM to select the most promising one (Yao et al., 2023; Besta et al., 2024). A central challenge in this setting is test-time compute optimality (TCO), which aims to select the optimal hyperparameters that maximize answer accuracy with a fixed policy model and given compute budget (Snell et al., 2024; Liu et al., 2025; Qu et al., 2025). Empirical studies suggest that among these components, the PRM often plays a critical role in determining TCO performance (Liu et al., 2025). However, most PRMs are trained via supervised learning on limited datasets (Lightman et al., 2023; Wang et al., 2024b), and their ability to generalize to unseen reasoning paths significantly affects the accuracy of path selection. Despite their growing importance, the effects of the generalization error of PRM on downstream reasoning performance remain underexplored.

To systematically characterize the role of PRMs in achieving TCO, we propose a unified theoretical framework. We first show that the generalization error of the PRM is upper-bounded via PAC-Bayesian analysis. Then, we establish the relationship between the generalization error, the final answer accuracy, and the available computing budget. One of our key insights is to quantify the risk of mis-ranking candidate reasoning paths due to generalization error. Our analysis shows that the answer accuracy of external TTS is lower-bounded by three components: **(i)** the probability that

the policy model generates a correct answer, **(ii)** the **reward gap** between the selected path and the discarded ones, and **(iii)** the upper bound of the generalization error.

While our theoretical framework provides a target for optimizing inference-time accuracy, it also reveals two key practical challenges. First, the generalization error of the PRM significantly affects answer accuracy under a fixed compute budget, yet it is unobservable at test time. Second, although the reward gap can be influenced by tuning sampling parameters such as top-$k$, top-$p$, and temperature, its effect varies across different PRMs. To address these challenges, we use parameter sparsity as a proxy for the generalization error of the PRM and propose Compute-Aware Tree Search (CATS), a dynamic reasoning control strategy based on the Advantage Actor-Critic (A2C) framework. CATS formulates the inference process as a Markov Decision Process (MDP) and controls the search configuration by learning an actor network. At each step, the actor network outputs a search configuration based on candidate rewards and model sparsity, while the critic estimates the value of the current state via a temporal-difference (TD) objective. By jointly training on multiple PRMs, CATS learns to adaptively adjust the number and selection of candidate reasoning paths, effectively improving the answer accuracy under limited compute.

To validate our theoretical analysis and the effectiveness of the proposed CATS strategy, we conduct extensive experiments on two challenging mathematical reasoning benchmarks: MATH500 (Hendrycks et al., 2021; Lightman et al., 2023) and AIME24 (AI-MO, 2024). We evaluate CATS under multiple frozen policy models, including Qwen 2.5 (Yang et al., 2024), Llama 3.1 (Grattafiori et al., 2024), and Llama 3.2 (Meta AI, 2024), and incorporate a diverse set of PRMs (Zhang et al., 2025b; Wang et al., 2024b; Dong et al., 2024; Skywork o1 Team, 2024). The results demonstrate that CATS consistently achieves higher accuracy than other external TTS methods across different model combinations and compute budgets. These results provide strong empirical support for our theoretical predictions. Our contributions can be summarized as follows:

- We present a unified theoretical framework that establishes a quantitative relationship between the generalization error of the PRM, compute budget, and answer accuracy in external TTS. By analyzing the risk of mis-ranking candidate reasoning paths, we derive an explicit lower bound on answer accuracy in terms of the reward gap, sampling coverage, and the generalization error.

- Motivated by our theoretical analysis, we propose CATS, a reasoning control strategy based on the A2C framework. CATS dynamically allocates compute across inference steps by adjusting path selection and generation parameters, using sparsity as a proxy signal for the generalization error of PRM.

- We evaluate CATS across diverse challenging reasoning benchmarks, different policy models, and several PRMs. Results show that CATS consistently improves accuracy across all settings, validating our theoretical predictions.

## 2 RELATED WORKS

**Scaling of Test-time Compute**. CoT prompting is first proposed as a prompting technique that enables LLMs to decompose problems into intermediate steps (Wei et al., 2022). Recently, the OpenAI o1 series (OpenAI, 2024) demonstrate that increasing the length of CoT during inference yields substantial performance gains on tasks like MATH (Hendrycks et al., 2021) and AIME (AI-MO, 2024). TTS approaches can be broadly categorized into internal and external methods (Snell et al., 2024; Liu et al., 2025; Chen et al., 2025). Internal TTS encourages models to extend CoT reasoning via supervised fine-tuning (SFT) or reinforcement learning (RL). Some methods construct training data to promote step-wise self-refinement (Madaan et al., 2023; Saunders et al., 2022). DeepSeek-R1 (DeepSeek-AI, 2025) combines formatting-based and rule-based rewards, and optimizes the model using GRPO (Shao et al., 2024). In contrast, external TTS improves the reasoning performance via sampling or search-based methods with fixed LLMs and an external verifier (Lightman et al., 2023; Snell et al., 2024; Sel et al., 2024; Besta et al., 2024). Specifically, Snell et al. (Snell et al., 2024) analyze compute-optimal test-time scaling, finding that adaptive allocation of verifier-guided search can beat a 14× larger model while using 4× less extra compute than best-of-N sampling.

**Process Reward Model**. An essential component of external TTS is the verifier that evaluates different reasoning paths. Verifiers are categorized into two types: Process Reward Models (PRMs)

and Outcome Reward Models (ORMs) (Uesato et al., 2022). PRMs assess the quality of a reasoning step given the question and partial reasoning trajectory, estimating the likelihood that the process will lead to a correct answer (Lightman et al., 2023; Wang et al., 2024b). In contrast, ORMs provide a reward signal based on the final answer's correctness, given the full reasoning trace and output (Uesato et al., 2022; Lightman et al., 2023). Recent studies have shown that PRMs are generally more effective than ORMs in guiding search (Lightman et al., 2023; Uesato et al., 2022; Snell et al., 2024), and PRMs have become a widely adopted tool in external test-time reasoning frameworks (Liu et al., 2025; Snell et al., 2024). Lightman et al (Lightman et al., 2023) trains PRM on 800k human-labeled reasoning steps. Math-Shepherd (Wang et al., 2024b) automatically constructs step-level supervision by forward decoding multiple reasoning branches from each intermediate step and assigning scores based on the proportion of branches that reach the known correct answer.

## 3 PROBLEM FORMULATION

In this section, we first formalize the reasoning task and briefly introduce external TTS methods. We highlight TCO as a central objective in this setting. We then introduce PRM as a key component of the external TTS framework and describe its training process. Finally, we present the central motivation of this work: understanding how the generalization performance of PRM affects TCO.

### 3.1 THE PROBLEM DEFINITION OF REASONING AND SCALING OF TEST-TIME COMPUTE

Given an input question $q \in \mathcal{Q}$, the reasoning problem can be presented as outputting an answer $a \in \mathcal{A}$ that matches the ground-truth answer $a^*(q) \in \mathcal{A}$, using a policy model $\pi_\theta$. Here, $\mathcal{Q}$ and $\mathcal{A}$ denote the sample spaces of questions and answers, respectively, and $\pi_\theta$ is a pre-trained LLM. To tackle this challenge, a promising direction is to scale the test-time compute by allocating more computational resources during answer generation. Specifically, the policy model generates a CoT reasoning path $h = (z_1, z_2, \ldots, z_T)$ in an autoregressive manner, where each step $z_t$ is sampled from $\pi_\theta(\cdot \mid q, z_1, \ldots, z_{t-1})$, followed by sampling the final answer $a \sim \pi_\theta(\cdot \mid z_1, \ldots, z_T)$. This class of methods is also referred to as TTS methods. Existing TTS methods can be broadly categorized into two types: internal TTS and external TTS (Snell et al., 2024; Liu et al., 2025). While internal TTS modifies $\pi_\theta$ via fine-tuning to encourage longer reasoning paths for complex problems (OpenAI, 2024; DeepSeek-AI, 2025), in this work, we primarily focus on the external TTS methods, which samples a collection of reasoning paths $\mathcal{H} = \{h_1, h_2, \ldots, h_N\}$ and employs an external PRM $R_\phi$ to score each path $h \in \mathcal{H}$, selecting the one with the highest reward. The key distinction is that internal TTS fine-tunes $\pi_\theta$ to generate a long CoT reasoning path, while external TTS doesn't require fine-tuning.

Here, we briefly introduce two representative external TTS methods. The first is *Best-of-N* sampling. Given a question $q$, $N$ independent and complete reasoning paths $\{h_i\}_{i=1}^N$ are sampled from $\pi_\theta(\cdot \mid q)$. Each candidate path $h_i$ is evaluated by the PRM $R_\phi : \mathcal{Q} \times \mathcal{H} \to [0, 1]$, and outputs a reward $r_i \in [0, 1]$. The path with the highest reward is selected for the final output. The second is *Beam Search*. Starting from the initial input $q$, $N$ candidate first steps $\{z_{1,i}\}_{i=1}^N$ are sampled from $\pi_\theta(\cdot \mid q)$. The PRM scores each first step, and the top $N/M$ highest-scoring steps are retained, where $M$ is the beam width. For each retained step, $M$ next steps are sampled to expand into a total of $N$ second-step paths. This procedure is repeated iteratively: at each step, paths are expanded, scored, filtered, and expanded again, until $N$ complete reasoning paths are produced. Finally, the PRM evaluates the complete paths, and the highest-scoring path is selected.

A key challenge of external TTS is how to scale compute optimally. That is, given a fixed compute budget $C$, how to select the optimal hyperparameter $\psi$ that maximizes the probability of producing the correct answer for a problem $q$. We follow (Snell et al., 2024) and formalize the objective as:

$$\psi_{q,a^*(q)}^*(C) = \arg\max_\psi (\mathbb{E}_{a \sim \text{Target}(\psi, C, q)}[\mathbb{1}_{(a = a^*(q))}]), \tag{1}$$

where $\mathbb{1}_{(a = a^*(q))}$ is the indicator function that equals 1 if the selected answer $a$ matches the ground-truth answer $a^*(q)$, and 0 otherwise. $\text{Target}(\psi, C, q)$ denotes the distribution over outputs induced by executing the reasoning process under hyperparameter $\psi$ and budget $C$ on question $q$, and $\psi_{q,a^*(q)}^*(C)$ represents the optimal TTS strategy. The hyperparameter configuration $\psi$ includes, but is not limited to: **(i)** the choice of TTS strategy, such as Best-of-N, Beam Search, or other chain-of-thought-based

methods; **(ii)** sampling parameters used during generation, such as top-$k$ truncation, top-$p$ truncation, and temperature; and **(iii)** for Beam Search, the beam width $M$ maintained at each reasoning step. The compute budget $C$ can be interpreted in multiple ways depending on the context. It can refer to the maximum number of tokens generated during test-time inference, or, as defined in (Snell et al., 2024), the total number of reasoning paths.

## 3.2 PRM: PROCESS REWARD MODEL

The PRM $R_\phi$ plays a central role in external TTS. It outputs a score for each reasoning step $z_t$ by inputting the reasoning prefix $h_t = (z_1, \ldots, z_t)$ and the input question $q$. A PRM is typically implemented by appending a linear prediction head to another LLM (different from $\pi_\theta$) and then fine-tuning the entire network on supervised training data (Uesato et al., 2022; Lightman et al., 2023; Zhang et al., 2025b; Wang et al., 2024b). The dataset $\mathcal{D} = \{(q_i, \{(h_{i,t}, y_{h_{i,t}})\}_{t=1}^{T_i})\}_{i=1}^n$ for training PRM consists of $n$ questions and corresponding reasoning steps $\{h_{i,t}\}_{t=1}^{T_i}$, where $q_i$ denotes the $i$-th input question, $h_{i,t} = (z_{i,1}, \ldots, z_{i,t})$ represents the reasoning prefix up to step $t$ for question $q_i$ and $y_{h_{i,t}} \in \{0, 1\}$ is the quality label for step $h_{i,t}$, with 1 indicating "good" and 0 indicating "bad". The label collection process can be found in (Lightman et al., 2023; Wang et al., 2024b). Given the above training data, the PRM training objective for each question $q_i$ is formulated as:

$$\mathcal{L}_{\text{PRM}} = \sum_{t=1}^{T_i} \Big( y_{h_{i,t}} \log r_{h_{i,t}} + (1 - y_{h_{i,t}}) \log(1 - r_{h_{i,t}}) \Big), \tag{2}$$

where $r_{h_{i,t}} = R_\phi(q_i, h_{i,t})$. After training, the parameters of PRM are frozen. Within the external TTS framework, the trained PRM is expected to assign reliable scores to the novel reasoning steps produced by the frozen policy $\pi_\theta$ when confronted with unseen questions.

## 3.3 MOTIVATION: THE RELATIONSHIP BETWEEN TCO AND PRM

Intuitively, the score of reasoning steps produced by the PRM directly influences the selection of reasoning paths, thereby can affect both the final answer accuracy and the computational consumption during inference. Since a PRM is usually obtained by fine-tuning an LLM on a limited set of training data, its prediction may generalize poorly to unseen questions. Therefore, investigating how such generalization ability of PRM affect TCO is critical for designing more efficient TTS methods. Specifically, we aim to address the following three key questions: **(i)** Under a fixed compute budget and a fixed reasoning strategy, how does the generalization ability of the PRM affect the accuracy of the final answer? **(ii)** Given the accuracy of the target answer, how does the generalization ability of the PRM influence the required compute budget? **(iii)** How can we dynamically allocate compute during inference based on the reward model's generalization behavior, to improve the final answer accuracy under a fixed total compute budget? The first two questions aim to characterize how the generalization ability of the PRM affects TCO, while the third question focuses on designing an external TTS method that achieves TCO by leveraging the theoretical insights.

## 4 THEORETICAL ANALYSIS

In this section, we develop a theoretical framework to answer the three questions above. We begin by modeling an upper bound on the generalization error of PRM within the PAC-Bayes framework. Next, to address the first question, we analyze how this bound affects answer accuracy through the risk of mis-ranking candidate paths. For the second question, we derive how this bound affects the compute budget required for a desired accuracy level. Finally, to address the third question, we examine how the theoretical findings inform the design of external TTS methods.

## 4.1 GENERALIZATION BOUNDS FOR REWARD MODELS

As discussed in Section 3.2, the PRM is typically trained using supervised learning on a limited set of annotated examples. However, at test time, the PRM must evaluate inputs that include not only previously unseen questions but also new reasoning paths generated by the policy model. In such cases, the PRM is still expected to assign reliable scores to candidate reasoning paths. We refer to

this ability as the generalization ability of PRM, which we identify as a key factor influencing both reasoning performance and computational efficiency.

Let $\mathcal{D}$ be an unknown data distribution over $\mathcal{Q} \times \mathcal{H} \times \mathcal{Y}$, where each data point consists of a question $q$, a reasoning path $h$, and a binary label $y$ indicating whether the path is helpful for solving the question. Let $\phi \in \Phi$ denote the parameters of PRM $R_\phi$, $\Phi$ is the parameter space. Let $\ell(R_\phi(q, h), y) = |R_\phi(q, h) - y| \in [0, 1]$ be the absolute error between the model's output to a ground-truth label $y \in \{0, 1\}$. Under the following assumption:

**Assumption 4.1.** *The data sample $(q, h, y)$ are drawn i.i.d. from the distribution $\mathcal{D}$.*

Let $\mathcal{L}_\mathcal{D}(\phi) = \mathbb{E}_{(q,h,y)\sim\mathcal{D}}[\ell(R_\phi(q, h), y)]$ be the population risk, and $\mathcal{L}_\mathcal{S}(\phi) = \frac{1}{n}\sum_{i=1}^{n}\ell(R_\phi(q_i, h_i), y_i)$ be the empirical risk on a training dataset $\mathcal{S} = \{(q_i, h_i, y_i)\}_{i=1}^{n}$. The generalization error $\varepsilon_{\text{gen}}(\phi)$ is defined as:

$$\varepsilon_{\text{gen}}(\phi) := \mathcal{L}_\mathcal{D}(\phi) - \mathcal{L}_\mathcal{S}(\phi) \tag{3}$$

It quantifies how much the model's performance on unseen data deviates from its performance on the training set $\mathcal{S}$. In practice, a PRM is typically trained to minimize the empirical loss on $\mathcal{S}$ (Lightman et al., 2023; Wang et al., 2024b), resulting in a small training loss. Thus, the generalization error captures the extent to which the predicted rewards deviate from the ground-truth rewards on unseen reasoning paths and questions.

We adopt the PAC-Bayes framework to analyze $\varepsilon_{\text{gen}}(\phi)$. Let $(\Phi, \mathcal{B})$ be the measurable space of model parameters $\phi$, where $\mathcal{B}$ is the Borel $\sigma$-algebra on $\Phi$. Let $\mathcal{P}(\Phi)$ denote the set of all probability measures over $(\Phi, \mathcal{B})$. A prior distribution $P \in \mathcal{P}(\Phi)$ is a probability distribution over model parameters $\phi$, which is the learner's initial assumption before seeing data. After seeing a training dataset $\mathcal{S} = \{(q_i, h_i, y_i)\}_{i=1}^{n} \sim \mathcal{D}^n$, the learner selects a posterior distribution $Q \in \mathcal{P}(\Phi)$ over parameters $\phi$, which is the learner's belief after observing the training set.

**Theorem 4.2** (PAC-Bayes Generalization Bound for PRMs)**.** *Let $P, Q \in \mathcal{P}(\Phi)$ be any prior and posterior distributions over $\phi$, and let $\ell$ be a bounded loss function taking values in $[0, 1]$. Then, for any $\delta \in (0, 1]$, with probability at least $1 - \delta$ over the choice of training set $\mathcal{S} \sim \mathcal{D}^n$, the following inequality holds:*

$$\mathbb{E}_{\phi\sim Q}[\mathcal{L}_\mathcal{D}(\phi)] \leq \mathbb{E}_{\phi\sim Q}[\mathcal{L}_\mathcal{S}(\phi)] + \sqrt{\frac{\text{KL}(Q\|P) + \log\frac{n}{\delta}}{2(n-1)}}. \tag{4}$$

The proof is provided in Appendix B. Equation 4 can be rewritten as the expected generalization error:

$$\mathbb{E}_{\phi\sim Q}[\varepsilon_{\text{gen}}(\phi)] \leq \sqrt{\frac{\text{KL}(Q\|P) + \log\frac{n}{\delta}}{2(n-1)}}. \tag{5}$$

Equation 5 shows that, with probability at least $1 - \delta$ over the draw of the training set $\mathcal{S}$, the expected deviation of the predicted reward from the true reward under the learner's belief $Q$ is upper-bounded by the sample size $n$ and the divergence $\text{KL}(Q\|P)$ between posterior and prior. In practice, the PRM is typically fixed after training, which corresponds to using a Dirac posterior $Q = \delta_{\hat{\phi}}$. In this case, the PAC-Bayes bound reduces to a pointwise guarantee $\varepsilon_{\text{gen}}(\hat{\phi}) \leq \sqrt{(\log(1/P(\hat{\phi})) + \log(n/\delta))/2(n-1)}$, and the KL term becomes $\log(1/P(\hat{\phi}))$, reflecting how well the learned parameters align with the prior. Next, we analyze how this bound influences the final answer accuracy and how it relates to the problem of TCO.

### 4.2 IMPACT OF REWARD MODEL GENERALIZATION ON ANSWER ACCURACY

In the external TTS framework, the PRM selects the path with the highest predicted score, and the answer is correct only if this selected path is also the truly highest-reward path. When the output of PRM is accurate, the top-scored path indeed has the highest true reward. When the predicted scores deviate from the true rewards, two cases arise: **(i)** the predicted top path still coincides with the true top path; or **(ii)** prediction error causes the true best path to be ranked lower and hence not selected. The second case may lead the system to choose a suboptimal path and thereby reduce answer accuracy.

To quantify this effect, we develop a theoretical framework that relates the generalization error of PRM to the accuracy of the selected answer.

Let $\mathcal{H} = \{h_1, h_2, \ldots, h_N\}$ denote the set of candidate reasoning paths independently sampled from the policy model $\pi_\theta(\cdot \mid q)$ for a given question $q$, i.e., $\mathcal{H} \sim \pi_\theta^{\otimes N}$. Here, $\pi_\theta^{\otimes N}$ denotes the joint distribution of $N$ independent samples from $\pi_\theta(\cdot|q)$. The goal of external TTS is to select the highest-scoring path $h_{\text{sel}}$ according to a learned PRM $R_\phi(q, h)$, i.e., $h_{\text{sel}} = \arg\max_{h \in \mathcal{H}} R_\phi(q, h)$, with the hope that the selected path yields the correct answer, i.e., $a(h_{\text{sel}}) = a^*(q)$. Therefore, assuming access to a ground-truth reward function $R^*(q, h) \in [0, 1]$, it is reasonable to assume that a reasoning path with a sufficiently high ground-truth reward should lead to a correct answer.

**Assumption 4.3** (Path-to-Answer Correctness). *There exists a threshold $\tau \in (0, 1]$ such that for any $h \in \mathcal{H}$, if $R^*(q, h) \geq \tau$, then $a(h, q) = a^*(q)$, where $a(h)$ denotes the final answer of path $h$.*

Furthermore, motivated by empirical observations (Brown et al., 2024), we assume that as the number of sampled paths increases, the probability that $\mathcal{H}$ contains at least one high-reward path approaches 1. Formally:

**Assumption 4.4** (Asymptotic Coverage). *Let $p_{N,\tau}(q) := \Pr_{\mathcal{H} \sim \pi_\theta^{\otimes N}} \left[ \exists h \in \mathcal{H}, \ R^*(q, h) \geq \tau \right]$. Then $\lim_{N \to \infty} p_N(q) = 1$.*

Under these assumptions, the only remaining source of error lies in the ranking behavior of the PRM: even if a high-quality path is present in the candidate set $\mathcal{H}$, the PRM may fail to rank it highest due to the generalization error, which we can upper-bounded according to Equation 5. Define this upper-bound as $\varepsilon$, we propose the following theorem.

**Theorem 4.5** (Answer–Accuracy Bound with Reward-Gap). *Let $q$ be a fixed question, and let $\mathcal{H} = \{h_1, \ldots, h_N\} \sim \pi_\theta^{\otimes N}$ be $N$ i.i.d. candidate reasoning paths. Define $h^* = \arg\max_{h \in \mathcal{H}} R^*(q, h)$, and $h_{\text{sel}} = \arg\max_{h \in \mathcal{H}} R_\phi(q, h)$. The reward-gap $\gamma(q) = R^*(q, h^*) - \max_{h \in \mathcal{H} \setminus \{h^*\}} R^*(q, h) \geq 0$. Let $p_{N,\tau}(q) = \Pr_{\mathcal{H} \sim \pi_\theta^{\otimes N}} \left[ \exists h \in \mathcal{H} : \ R^*(q, h) \geq \tau \right]$. Suppose further that the following hold:*

- *There exist $\varepsilon \in (0, 1]$ and $\delta \in (0, 1)$ such that $\Pr \left[ \sup_{h \in \mathcal{H}} \left| R_\phi(q, h) - R^*(q, h) \right| \leq \varepsilon \right] \geq 1 - \delta$. And denote the high-probability event by $\mathcal{G} = \{\sup_h |R_\phi - R^*| \leq \varepsilon\}$.*

- *Conditioned on $\mathcal{H}$, the deviations $\Delta_h = R_\phi(q, h) - R^*(q, h)$ are independent, mean-zero, and satisfy $|\Delta_h| \leq \varepsilon$ almost surely.*

- *Assumptions 4.3 and Assumption 4.4 hold.*

*Then the probability of selecting a correct answer satisfies*

$$\Pr \left[ a(h_{\text{sel}}) = a^*(q) \right] \geq p_{N,\tau}(q) \left[ 1 - \delta - (N - 1) \exp\left( -\frac{\gamma(q)^2}{8\varepsilon^2} \right) \right]. \tag{6}$$

The proof is provided in Appendix C. Theorem 4.5 shows that the answer accuracy of external TTS is lower-bounded by $p_N(q)$ and $\exp(-\gamma(q)^2/(8\varepsilon^2))$, where $p_N(q)$ reflects the chance of sampling at least one high-quality path under fixed budget $N$, and $\varepsilon$ reflect the generalization error of PRM. This implies that the answer accuracy depends jointly on the sampling ability of the policy model and the generalization error of the PRM.

### 4.3 IMPACT OF REWARD MODEL GENERALIZATION ON COMPUTE BUDGET

Next, based on Theorem 4.5, we propose the following corollary to describe the budget requirement as a function of the upper-bound of the generalization error of PRM $\varepsilon$ and reward gap $\gamma(q)$.

**Corollary 4.6** (Target Accuracy Constraint on Sampling and Margin). *Given a generalization error bound of PRM $\varepsilon > 0$, a confidence parameter $\delta \in (0, 1)$, and a target answer accuracy level $\alpha \in (0, 1)$. Under the assumptions of Theorem 4.5, if one wishes to guarantee $\Pr[a(h_{\text{sel}}) = a^*(q)] \geq \alpha$, then the sampling coverage probability must satisfy*

$$p_{N,\tau}(q) \geq \frac{\alpha}{1 - \delta - (N-1) \exp(-\gamma(q)^2 / 8\varepsilon^2)}. \tag{7}$$

This corollary shows that, for higher generalization error of PRM, more reasoning paths need to be sampled to guarantee a higher accuracy $\alpha$. Therefore, the generalization ability of the PRM also significantly affects the compute budget, which is used to achieve a higher accuracy.

### 4.4 Inspiration for Designing External TTS Methods

The analysis in Section 4.2 and Section 4.3 addresses the first two problems we propose. For the third problem, Theorem 4.5 and Corollary 4.6 provide direct insight into the core objective of TCO as in Equation 1: selecting search hyperparameters that maximize answer accuracy under a fixed compute budget. Specifically, Equation 6 shows that answer accuracy increases with larger reward margin $\gamma(q)$, while it is negatively impacted by the generalization error bound $\varepsilon$ of the PRM. Although $\varepsilon$ is unknown at test time, the reward gap $\gamma(q)$ can be influenced by the sampling parameters, such as top-$k$, top-$p$, and temperature, which can affect the diversity of candidate reasoning paths generated by the policy model. This, in turn, modifies the $\gamma(q)$ among candidates. However, several challenges arise in practice: **(i)** different PRMs may have different generalization behaviors, requiring different reward margins $\gamma(q)$ to ensure reliable selection; **(ii)** generating candidate paths that satisfy a desired reward margin may require multiple sampling rounds. These observations motivate the design of a dynamic control mechanism that generates paths based on the generalization behaviors of PRMs.

## 5 Methodology

As we discussed in Section 4.4, an effective inference-time strategy should **(i)** be aware of the generalization behavior of the reward model, and **(ii)** increase $\gamma(q)$ dynamically without inducing additional compute cost. To address the first problem, we propose using the structural sparsity of $\phi$ as a proxy for estimating its generalization capacity. To address the second problem, we propose *Compute-Aware Tree Search* (CATS), a dynamic compute allocation framework based on A2C framework. During inference, the actor observes the current reasoning state and outputs search hyperparameters. The critic estimates the utility of each action and provides feedback to optimize the actor via reinforcement learning. This design allows CATS to adaptively adjust computation at each reasoning step while maintaining a global compute budget.

### 5.1 Estimation of $\varepsilon$ via Sparsity

In practice, the true generalization error $\varepsilon_{\text{gen}}(\hat{\phi})$ in Theorem 4.2 is unobservable and its PAC-Bayes upper bound depends on the prior density $P(\hat{\phi})$, which is rarely known in closed form. However, under structural assumptions on $\hat{\phi}$, we can approximate $\log(1/P(\hat{\phi}))$ using model-dependent statistics as a proxy. One common and well-motivated assumption is parameter sparsity, which reflects the idea that only a small subset of model parameters are relevant for capturing the reward signal. Sparsity-based priors have been widely used in PAC-Bayesian analysis to obtain non-vacuous generalization bounds (Muthukumar & Sulam, 2023; Lotfi et al., 2022), and have also proven effective in various practical settings (Roy et al., 2021; Jiang et al., 2024a). Under sparsity-based priors, the KL-divergence term can be upper-bounded by a function of the number of nonzero parameters in $\hat{\phi}$. This yields the following sparsity-induced bound on the generalization error:

$$\varepsilon_{\text{gen}}(\hat{\phi}) \ \leq \ \sqrt{\frac{c \cdot \|\hat{\phi}\|_0 \cdot \log d + \log \frac{n}{\delta}}{2(n-1)}}. \tag{8}$$

This expression provides a practical surrogate: models with fewer active parameters are expected to generalize better. We provide empirical evidence in Appendix H.

### 5.2 Compute-Aware Tree Search

To dynamically allocate compute budget at each reasoning step based on the generalization behavior of the reward model, we propose Compute-Aware Tree Search (CATS). In this approach, we formalize the reasoning process as a Markov Decision Process (MDP). Formally, we define the reasoning control problem as $(\mathcal{S}, \mathcal{A}, P, r, \gamma)$. The state space $\mathcal{S}$ captures the current search context, including: the number of candidate paths at the current step, their associated reward scores, parameter sparsity of the

reward model, and the maximum candidate paths that can be sampled. The action space $\mathcal{A}$ consists of a set of search hyperparameter configurations, including the number of additional candidates to sample, the number of candidates to retain for the next step, and the sampling parameters (e.g., top-$p$, top-$k$, and temperature). The transition function $P$ is deterministic: the next state is determined by applying the chosen action, either by sampling additional candidates and then retaining a subset, or by directly retaining a subset of existing candidates to advance to the next reasoning step. The reward function $r(s_t, a_t)$ is defined as follows:

$$r(s_t, a_t) = -\lambda_c \cdot C(a_t) + \lambda_m \cdot \Delta_m(s_t, a_t) + \lambda_r \cdot \max_{h \in \mathcal{H}} R_\phi(q, h), \tag{9}$$

where $C(a_t)$ denotes the additional candidate path incurred by action $a_t$, $\Delta_m(s_t, a_t)$ denotes the reward gap between the retained paths and discarded paths, $\max_{h \in \mathcal{H}} R_\phi(q, h)$ is the highest score of the candidate paths, $\lambda_c, \lambda_m, \lambda_r$ are hyperparameters. These rewards encourage high-quality generations and mitigate the risk of mis-pruning good paths. And $\gamma \in (0, 1]$ is a discount factor. Under this formulation, the objective of CATS is to learn a control policy $\pi_\nu(a_t \mid s_t)$ that maximizes the expected return throughout the reasoning process. Further details are in Appendix G

We employ an A2C framework (Sutton et al., 1999) to optimize the tree expansion policy via single-step temporal difference (TD) learning. The actor network $\pi_\nu(a_t \mid s_t)$ is based on a multi-layer perceptron (MLP) that outputs action probability. The critic network $V_\xi(s_t)$ is implemented as a separate MLP that predicts the scalar value of a given state. At each search step, the agent collects transition tuples $(s_t, a_t, r_t, s_{t+1})$ and computes the TD error:

$$\delta_t = r_t + \gamma V_\xi(s_{t+1}) - V_\xi(s_t), \tag{10}$$

which serves both as a regression target for the critic and as an advantage estimate for the actor. The critic is trained to minimize the squared TD error, while the actor is trained to maximize the expected return using the advantage-weighted log-probability objective:

$$\mathcal{L}_{\text{critic}}(\xi) = \frac{1}{2} (\delta_t)^2, \mathcal{L}_{\text{actor}}(\nu) = -\log \pi_\nu(a_t \mid s_t) \cdot \delta_t. \tag{11}$$

Gradients are computed with respect to $\xi$ and $\nu$, and updates are applied after each environment step. By optimizing Equation 11, the actor learns to produce actions at each step that maximize reward. During the testing phase, the actor can generate candidate reasoning paths with higher PRM scores and larger reward gaps, which helps prevent mis-ranking and ultimately improves answer accuracy. The pseudo codes for training and using CATS are provided in Appendix J.

## 6 EXPERIMENTS

### 6.1 IMPLEMENTATION DETAILS

To train the actor and critic networks, we follow the procedure in (Lightman et al., 2023) and construct a training set using 12,000 examples from the MATH dataset (Hendrycks et al., 2021). Each training sample consists of a question $q$ and its corresponding ground-truth answer $a^*(q)$. During training, we fix a policy LLM to generate candidate answers and collect data by scoring the reasoning paths under different PRMs. These trajectories are then used to train both the actor and critic networks. The actor network is implemented as a two-layer MLP with a hidden dimension of $128$ while the critic network is also implemented as a two-layer MLP with a hidden dimension of $256$. The hyperparameters $\lambda_c = 0.2, \lambda_m = 0.5, \lambda_r = 0.3$. We use the Adam optimizer with a learning rate of $1 \times 10^{-3}$ and train the models under different compute budgets. The ablation study is in Appendix I.

### 6.2 EXPERIMENTAL SETUP

We evaluate the proposed CATS method on two mathematical reasoning benchmarks: MATH-500 (Lightman et al., 2023) and AIME24 (AI-MO, 2024). To assess the generality of our approach, we evaluate CATS across a diverse set of frozen policy models, including LLaMA3.1-Instruct-8B, LLaMA3.2-Instruct-1B, Qwen2.5-Instruct (0.5B, 3B, and 7B). For the reward models, we include Math-Shepherd-PRM-7B, RLHFlow-PRM-Mistral-8B, RLHFlow-PRM-DeepSeek-8B, Skywork-PRM-1.5B, and Qwen2.5-Math-PRM-7B. The maximum number of candidate reasoning paths is 256. We compare CATS with three external TTS methods: Best-of-N, Beam Search with width $M = 4$,

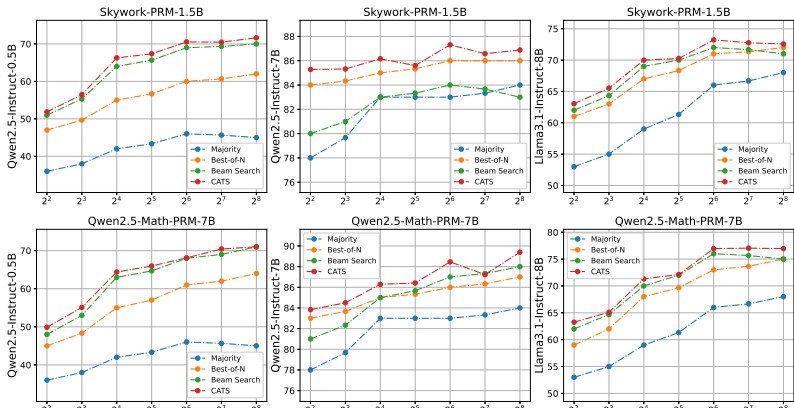

Figure 1: The comparison results on the MATH-500 dataset for different policy models and PRMs.

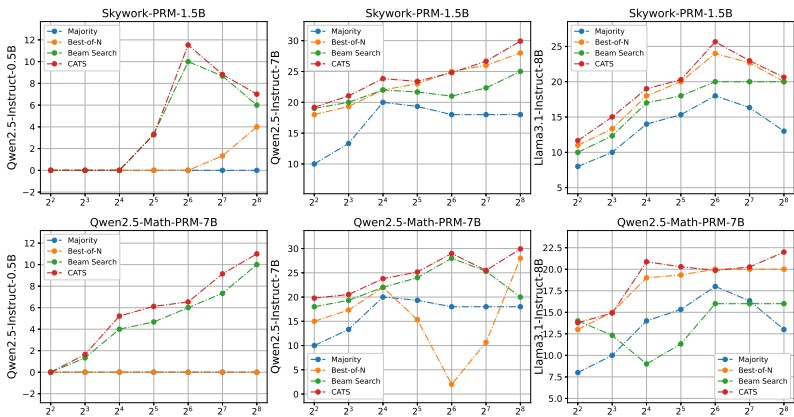

Figure 2: The comparison results on the AIME24 dataset for different policy models and PRMs.

and Majority Voting. For each method, we report answer accuracy as a function of the number of candidate paths, using $N \in \{4, 8, 16, 32, 64, 128, 256\}$. The full results are provided in Appendix F. And we also present the comparison results with other external TTS methods in Appendix E.

### 6.3 RESULTS

The performance of Qwen2.5-Instruct-0.5B, Qwen2.5-Instruct-7B, and LLaMA3.1-Instruct-8B on the MATH-500 dataset, evaluated under two PRMs: Skywork-PRM-1.5B and Qwen2.5-Math-PRM-7B, is shown in Figure 1. From the results, we observe that for different compute budgets and PRMs, the best-performing baseline (excluding CATS) varies. In contrast, CATS consistently outperforms all baselines across all budget levels and PRMs. The results on the AIME24 dataset are shown in Figure 2. Although AIME24 poses greater challenges than MATH-500, CATS continues to outperform other external TTS methods. These findings confirm the effectiveness of our proposed approach.

### 7 CONCLUSION

In this work, we analyze how the generalization error of the PRM influences the performance of the external TTS. By quantifying the mis-ranking risk induced by reward prediction error, we derive an explicit lower bound involving the reward gap and path coverage probability, which motivates the need for adaptive control over reasoning computation. Building on this insight, we propose CATS, which learns to allocate compute based on PRM proxies and effectively balances reward separation and candidate diversity. Extensive experiments on MATH and AIME24 demonstrate that CATS consistently outperforms standard search strategies across a wide range of policy models and PRMs.

ETHICS STATEMENT

All authors have carefully read and fully adhered to the ICLR Code of Ethics. The research presented in this paper was conducted in compliance with the principles of research integrity, fairness, and transparency. Our work does not involve human subjects, personally identifiable information, or other sensitive data, and it does not present foreseeable risks of harm, misuse, or ethical concerns. All experiments are carried out on publicly available datasets or synthetic data, and we have ensured proper documentation and reproducibility. The authors confirm that there are no conflicts of interest or violations of the ICLR Code of Ethics associated with this submission.

REPRODUCIBILITY STATEMENT

We ensure reproducibility by detailing experimental settings, datasets, and hyperparameters in both Section 6 and Appendices I. The source code is included in the supplementary materials to reproduce the proposed method.

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

APPENDIX

# A  LIMITATIONS

Our theoretical results are based on a set of reasonable assumptions, which may not fully hold in practical scenarios. However, these assumptions do not undermine the validity of the analysis, and our empirical results support the overall conclusions.

# B  PROOF OF THEOREM 4.2

**Theorem B.1** (PAC-Bayes Generalization Bound for Reward Models). *Let $P, Q \in \mathcal{P}(\Phi)$ be any prior and posterior distributions over reward model parameters, and let $\ell$ be a bounded loss function taking values in $[0, 1]$. Then, for any $\delta \in (0, 1]$, with probability at least $1 - \delta$ over the choice of training set $\mathcal{S} \sim \mathcal{D}^n$, the following inequality holds:*

$$\mathbb{E}_{\phi \sim Q}\left[\mathcal{L}_{\mathcal{D}}(\phi)\right] \leq \mathbb{E}_{\phi \sim Q}\left[\mathcal{L}_{\mathcal{S}}(\phi)\right] + \sqrt{\frac{\mathrm{KL}(Q\|P) + \log\frac{n}{\delta}}{2(n-1)}}. \tag{12}$$

*Proof.* The subsequent proof follows the classical PAC-Bayes derivation (McAllester, 1999; Seeger, 2002) and we prove it again in our scenario. Throughout, we work on the probability space induced by the i.i.d. sample $\mathcal{S} \sim \mathcal{D}^n$ (Assumption 4.1).

For any measurable function $f \colon \Phi \to \mathbb{R}$ and any posterior $Q \ll P$, the *Donsker–Varadhan* variational formula yields:

$$\mathbb{E}_{\phi \sim Q}[f(\phi)] \leq \frac{1}{\lambda}\left(\log \mathbb{E}_{\phi \sim P}\left[e^{\lambda f(\phi)}\right] + \mathrm{KL}(Q\|P)\right), \quad \forall \lambda > 0. \tag{13}$$

Fix $\phi \in \Phi$ and let

$$Z_i := \ell(R_\phi(q_i, h_i), y_i) \in [0, 1] \tag{14}$$

for $i = 1, \dots, n$. By Hoeffding's inequality,

$$\mathbb{E}_{\mathcal{S}}\left[\exp\left(\lambda(\mathcal{L}_{\mathcal{D}}(\phi) - \mathcal{L}_{\mathcal{S}}(\phi))\right)\right] \leq \exp\left(\frac{\lambda^2}{8n}\right), \qquad \forall \lambda \in \mathbb{R}. \tag{15}$$

Taking expectations over $\phi \sim P$ and applying Fubini's theorem gives

$$\mathbb{E}_{\mathcal{S}}\left[\mathbb{E}_{\phi \sim P}\left[e^{\lambda(\mathcal{L}_{\mathcal{D}}(\phi) - \mathcal{L}_{\mathcal{S}}(\phi))}\right]\right] \leq \exp\left(\frac{\lambda^2}{8n}\right). \tag{16}$$

Define the random variable

$$\Psi(\mathcal{S}) := \log \mathbb{E}_{\phi \sim P}\left[e^{\lambda(\mathcal{L}_{\mathcal{D}}(\phi) - \mathcal{L}_{\mathcal{S}}(\phi))}\right] - \frac{\lambda^2}{8n}. \tag{17}$$

By Equation 16, $\mathbb{E}_{\mathcal{S}}[\exp(\Psi(\mathcal{S}))] \leq 1$. Hence, by Markov's inequality,

$$\Pr_{\mathcal{S}}\left[\Psi(\mathcal{S}) > \log\frac{1}{\delta}\right] \leq \delta. \tag{18}$$

Thus, with probability at least $1 - \delta$ over $\mathcal{S} \sim \mathcal{D}^n$,

$$\log \mathbb{E}_{\phi \sim P}\left[e^{\lambda(\mathcal{L}_{\mathcal{D}}(\phi) - \mathcal{L}_{\mathcal{S}}(\phi))}\right] \leq \frac{\lambda^2}{8n} + \log\frac{1}{\delta}. \tag{19}$$

Condition on any $\mathcal{S}$ satisfying Equation 19. Applying Equation 13 with

$$f(\phi) = \lambda(\mathcal{L}_{\mathcal{D}}(\phi) - \mathcal{L}_{\mathcal{S}}(\phi)) \tag{20}$$

and the bound in Equation 19 yields

$$\lambda \mathbb{E}_{\phi \sim Q}\left[\mathcal{L}_{\mathcal{D}}(\phi) - \mathcal{L}_{\mathcal{S}}(\phi)\right] \leq \frac{\lambda^2}{8n} + \mathrm{KL}(Q\|P) + \log\frac{1}{\delta}. \tag{21}$$

Dividing by $\lambda > 0$ and optimizing w.r.t. $\lambda$ gives the tightest (sub-Gaussian) bound at

$$\lambda^\star = 4 \left(\frac{n-1}{2}\right)^{1/2}. \tag{22}$$

Plugging $\lambda^\star$ back leads to

$$\mathbb{E}_{\phi \sim Q}[\mathcal{L}_{\mathcal{D}}(\phi)] \leq \mathbb{E}_{\phi \sim Q}[\mathcal{L}_{\mathcal{S}}(\phi)] + \sqrt{\frac{\mathrm{KL}(Q\|P) + \log\frac{n}{\delta}}{2(n-1)}}. \tag{23}$$

Since the derivation holds on the $1 - \delta$ event triggered in Equation 19, the bound is valid with the claimed confidence level, which completes the proof. □

## C  PROOF OF THEOREM 4.5

**Theorem C.1** (Answer–Accuracy Bound with Reward-Gap Parameter). *Let $q$ be a fixed question, and let $\mathcal{H} = \{h_1, \ldots, h_N\} \sim \pi_\theta^{\otimes N}$ be $N$ i.i.d. candidate reasoning paths. Define $h^* = \arg\max_{h \in \mathcal{H}} R^*(q, h)$, and $h_{\mathrm{sel}} = \arg\max_{h \in \mathcal{H}} R_\phi(q, h)$. The reward-gap $\gamma(q) = R^*(q, h^*) - \max_{h \in \mathcal{H} \setminus \{h^*\}} R^*(q, h) \geq 0$. Let $p_{N,\tau}(q) = \Pr_{\mathcal{H} \sim \pi_\theta^{\otimes N}}\left[\exists h \in \mathcal{H} : R^*(q, h) \geq \tau\right]$. Suppose further that the following hold:*

- *There exist $\varepsilon \in (0, 1]$ and $\delta \in (0, 1)$ such that $\Pr\left[\sup_{h \in \mathcal{H}}\left|R_\phi(q, h) - R^*(q, h)\right| \leq \varepsilon\right] \geq 1 - \delta$. And denote the high-probability event by $\mathcal{G} = \{\sup_h |R_\phi - R^*| \leq \varepsilon\}$.*

- *Conditioned on $\mathcal{H}$, the deviations $\Delta_h = R_\phi(q, h) - R^*(q, h)$ are independent, mean-zero, and satisfy $|\Delta_h| \leq \varepsilon$ almost surely.*

- *Assumptions 4.3 and Assumption 4.4 hold.*

*Then the probability of selecting a correct answer satisfies*

$$\Pr\left[a(h_{\mathrm{sel}}) = a^*(q)\right] \geq p_{N,\tau}(q)\left[1 - \delta - (N-1)\exp\left(-\frac{\gamma(q)^2}{8\varepsilon^2}\right)\right]. \tag{24}$$

*Proof.* Define the failure events

$$E_1 = \left\{\mathcal{H} \cap \{h : R^*(q, h) \geq \tau\} = \varnothing\right\}, \quad E_2 = \{h_{\mathrm{sel}} \neq h^*\}. \tag{25}$$

By soft-correctness, success $\{a(h_{\mathrm{sel}}) = a^*(q)\}$ is the complement of $E_1 \cup E_2$. We bound:

$$\Pr(E_1) = 1 - p_{N,\tau}(q). \tag{26}$$

Condition on $E_1^c$ so that $h^*$ exists. On the event $\mathcal{G}$ we have

$$R_\phi(q, h_{\mathrm{sel}}) \geq R_\phi(q, h^*) \implies R^*(q, h_{\mathrm{sel}}) \geq R^*(q, h^*) - 2\varepsilon. \tag{27}$$

Hence any competitor $h \neq h^*$ must overcome a gap of at least $\gamma(q) - 2\varepsilon$. By Hoeffding's inequality for the bounded, independent deviations $\{\Delta_h\}_{h \neq h^*}$,

$$\Pr\left[E_2 \mid E_1^c, \mathcal{G}\right] \leq (N-1)\exp\left(-\frac{\gamma(q)^2}{8\varepsilon^2}\right). \tag{28}$$

Finally, applying the law of total probability and the union bound gives

$$\Pr(E_1 \cup E_2) \leq \Pr(E_1) + \Pr(E_2 \cap \mathcal{G} \mid E_1^c)\Pr(E_1^c) + \Pr(\mathcal{G}^c)$$

$$\leq (1 - p_{N,\tau}(q)) + p_{N,\tau}(q)(N-1)e^{-\gamma(q)^2/8\varepsilon^2} + \delta.$$

Subtracting from 1 yields the bound Equation 6. The asymptotic form for $\varepsilon/\gamma(q) \to 0$ follows by observing that $\exp(-\gamma(q)^2/8\varepsilon^2) \to 0$. □

# D  PROOF OF COROLLARY 4.6

**Corollary D.1** (Target Accuracy Constraint on Sampling and Margin). *Given a generalization error bound $\varepsilon > 0$, a confidence parameter $\delta \in (0, 1)$, and a target answer accuracy level $\alpha \in (0, 1)$. Under the assumptions of Theorem 4.5, if one wishes to guarantee $\Pr[a(h_{sel}) = a^*(q)] \geq \alpha$, then the sampling coverage probability must satisfy*

$$p_{N,\tau}(q) \geq \frac{\alpha}{1 - \delta - (N-1) \exp(-\gamma(q)^2 / 8\varepsilon^2)}. \tag{29}$$

*Proof.* By invoking Theorem 4.5 we have

$$\Pr\big[a(h_{sel}) = a^*(q)\big] \geq p_{N,\tau}(q) \left[1 - \delta - (N-1) \exp\left(-\frac{\gamma(q)^2}{8\varepsilon^2}\right)\right].$$

In order to guarantee $\Pr[a(h_{sel}) = a^*(q)] \geq \alpha$, it suffices to enforce

$$p_{N,\tau}(q) \left[1 - \delta - (N-1)e^{-\gamma(q)^2/(8\varepsilon^2)}\right] \geq \alpha. \tag{30}$$

Under the standing assumption that $1 - \delta - (N-1) \exp(-\gamma(q)^2/(8\varepsilon^2)) > 0$, we may divide both sides by this positive quantity, yielding

$$p_{N,\tau}(q) \geq \frac{\alpha}{1 - \delta - (N-1) \exp\big(-\gamma(q)^2/(8\varepsilon^2)\big)}, \tag{31}$$

which is precisely the bound stated in Equation 7. $\qquad\square$

# E  COMPARISON WITH OTHER METHODS

In this section, we present a comparison of answer accuracy between CATS and several recent external TTS baselines across three benchmark datasets: GSM8K (Cobbe et al., 2021), MATH, and OlympiadBench (He et al., 2024) in Table 1. The results show that CATS consistently outperforms

Table 1: Comparison of answer accuracy with other external TTS methods on the GSM8K, MATH, and OlympiadBench datasets.

| Method | Base Model | GSM8K | Math | OlympiadBench |
|---|---|---|---|---|
| STILL-1 (Jiang et al., 2024b) | Llama-3-8B-Instruct | - | - | 34.3 |
| LiteSearch (Wang et al., 2024a) | Llama-3-8B-Instruct | 75.7 | - | - |
| AlphaMath (Chen et al., 2024) | DeepSeekMath-7B-Base | 83.2 | 64.0 | - |
| MCTS-DPO (Xie et al., 2024) | Llama-3.1-8B-Instruct | 85.7 | - | - |
| NuminaMath-72B-CoT (Li et al., 2024) | Qwen2-72B | 90.8 | 66.7 | 32.6 |
| LLaMA-Berry (Zhang et al., 2024b) | Llama-3.1-8B-Instruct | 96.1 | 75.3 | 55.1 |
| MCTSr (Zhang et al., 2024a) | Llama-3-8B-Instruct | 96.7 | 58.2 | - |
| BoostStep (Zhang et al., 2025a) | Qwen2.5-Math-72B-Instruct | - | 85.2 | 52.7 |
| CATS | Llama-3.1-8B-Instruct | 97.1 | 76.9 | 56.1 |
| CATS | Llama-3.2-1B-Instruct | 88.4 | 61.8 | 33.6 |
| CATS | Qwen2.5-Instruct-3B | 96.5 | 79.3 | 38.1 |
| CATS | Qwen2.5-Instruct-7B | **98.0** | **89.4** | **58.4** |

prior methods under comparable base models. Notably, CATS surpasses methods based on much larger models such as Qwen2-72B and DeepSeekMath-72B. For instance, on GSM8K, CATS with Qwen2.5-7B attains an accuracy of 98.0%, exceeding the previous best result of 96.7% reported by MCTSr. These results demonstrate the effectiveness of our proposed CATS, even with smaller model sizes.

# F  FULL RESULTS

The full results of all policy models and PRMs in the MATH-500 dataset are provided in Figure 3, and the results of AIME are provided in Figure 4.

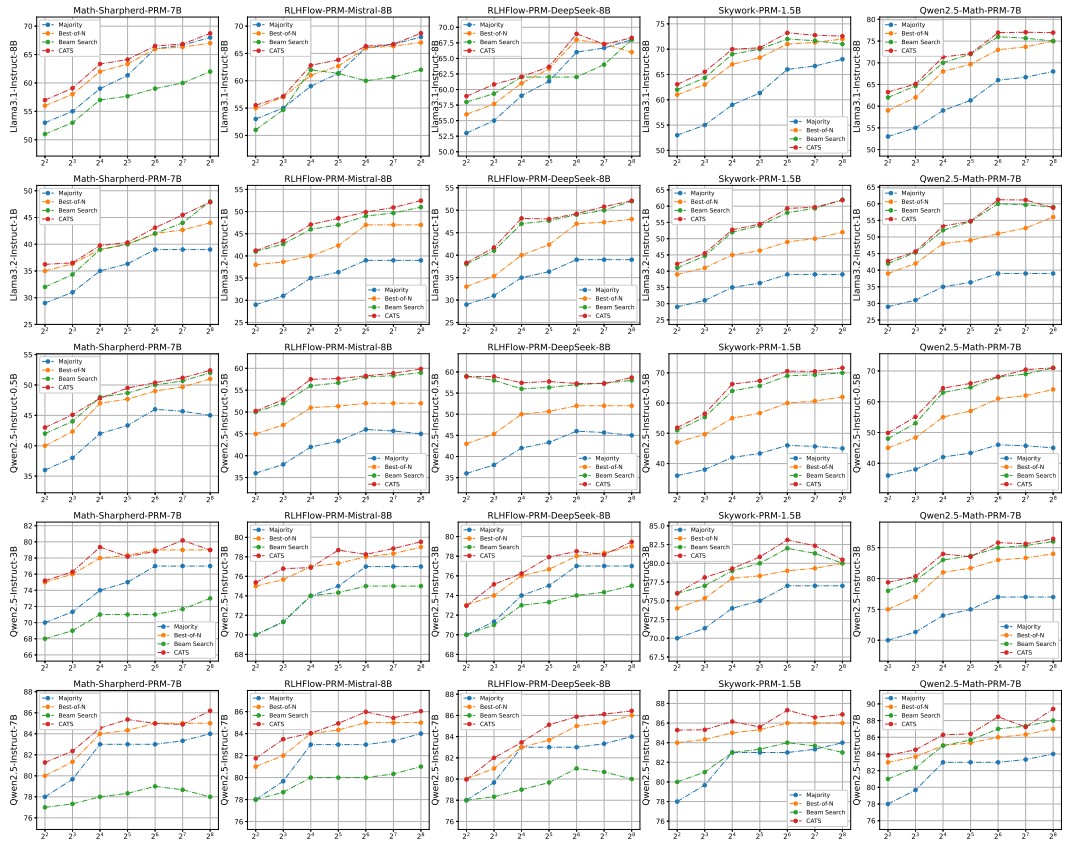

Figure 3: Full results on the MATH500 dataset.

# G IMPLEMENTATION DETAILS

## G.1 STATE REPRESENTATION

Both the actor and critic networks take as input a state vector $s_t$ defined as:

$$s_t = \left[n_{\text{init}}, \text{Top–}K(R), \max R, \text{mean } R, \text{std } R, \|\phi\|_0/d, \text{rem}\right] \tag{32}$$

Where:

- $R$: the PRM scores.
- $n_{\text{init}}$: the base expansion count per parent node (2 when $N = 4$, else 4).
- Top-$K$: the top-$K$ PRM scores from the current candidate set, with $K$ being a hyperparameter.
- $\max R, \text{mean } R, \text{std } R$: max, mean, and standard deviation of the PRM scores.
- $\|\phi\|_0/d$: sparsity ratio of the PRM parameters.
- rem: remaining candidate budget, decremented with each new full-path generation.

## G.2 ACTION SPACE

The action vector $a_t$ is defined as:

$$a_t = [\text{top-}k, \text{top-}p, \tau, n_{\text{sam}}, n_{\text{retain}}] \tag{33}$$

Where:

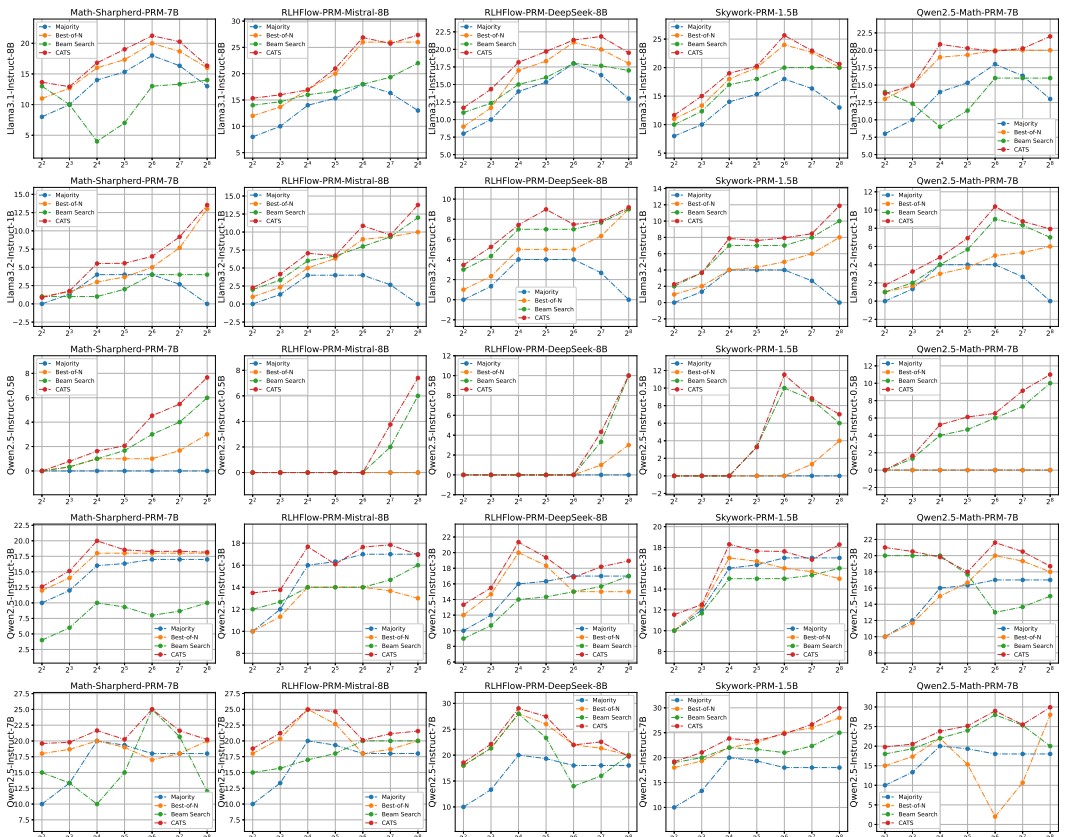

Figure 4: Full results on the AIME24 dataset.

- top-$k$: number of tokens retained before nucleus filtering.
- top-$p$: nucleus probability mass threshold.
- $\tau$: temperature for softmax sampling.
- $n_{\text{sam}}$: number of candidate paths to sample at the current step.
- $n_{\text{retain}}$: number of highest-scoring paths to keep.

### G.3 REWARD GAP

For state $s_t$ and action $a_t$, let $\text{retain}(a_t)$ and $\text{discard}(a_t)$ denote the retained and pruned paths, respectively. The margin is defined as:

$$\Delta_m(s_t, a_t) = \max_{h \in \text{retain}(a_t)} R_\phi(q, h) - \max_{h \in \text{discard}(a_t)} R_\phi(q, h) \tag{34}$$

This captures the PRM score difference between the best retained and best discarded paths. If no paths are pruned, $\Delta_m = 0$.

## H ANALYSIS OF SPARSITY IN PRM

In this section, we evaluate the validity of using sparsity as a proxy for reward model generalization error. Specifically, we consider two sparsity-based indicators: the overall parameter sparsity of the model and the sparsity of its output layer. The sparsity is calculated by counting the ratio of parameters with values smaller than $1 \times 10^{-4}$. To assess generalization performance, we use the test set from the PRM800K dataset, where each example contains a question, a reasoning step, and a binary label indicating the correctness of that step. For each reward model, we compute the

cross-entropy between its predicted reward scores and the ground-truth labels across the test set. This deviation reflects the degree of misalignment between predicted and true rewards, and thus serves as an empirical estimate of generalization error. The results is illustrated in Table 2. As presented

Table 2: The relationship between sparsity of different PRMs and the test error.

| PRM | #Params | Total Sparsity | Last layer Sparsity | Test Error |
|---|---|---|---|---|
| Math-Shepherd-PRM-7B | 7.11B | 0.0290 | 0.0196 | 2.78 |
| RLHFlow-PRM-Mistral-8B | 8.03B | 0.0068 | 0.0060 | 3.87 |
| RLHFlow-PRM-DeepSeek-8B | 8.03B | 0.0068 | 0.0060 | 3.87 |
| Skywork-PRM-1.5B | 1.54B | 0.0029 | 0.0059 | 4.43 |
| Qwen2.5-Math-PRM-7B | 7.08B | 0.0060 | 0.0080 | 3.47 |

in Table 2, we can observe a clear correlation between model sparsity and generalization behavior, supporting the use of sparsity as a practical and observable proxy in our control framework.

# I ABLATION STUDY

## I.1 THE ROLE OF PARAMETER SPARSITY IN CATS

To investigate the impact of parameter sparsity in the CATS framework, we conduct an ablation study comparing model performance with and without the parameter sparsity included in the state representation. Specifically, we use Qwen2.5-Instruct-7B as the policy model and evaluate the average performance on the MATH-500 and AIME24 datasets. The results are illustrated in Table 3. The

Table 3: Average accuracy on MATH-500 and AIME24 using Qwen2.5-Instruct-7B, with and without parameter sparsity as part of the state.

| PRM | MATH-500 | AIME-24 |
|---|---|---|
| *With Parameter Sparsity* | | |
| Math-Shepherd-PRM-7B | 83.64 | 20.70 |
| RLHFlow-PRM-Mistral-8B | 84.05 | 21.13 |
| RLHFlow-PRM-DeepSeek-8B | 83.61 | 22.27 |
| Skywork-PRM-1.5B | 85.98 | 23.55 |
| Qwen2.5-Math-PRM-7B | 86.26 | 24.09 |
| *Without Parameter Sparsity* | | |
| Math-Shepherd-PRM-7B | 82.14 | 20.40 |
| RLHFlow-PRM-Mistral-8B | 83.26 | 20.76 |
| RLHFlow-PRM-DeepSeek-8B | 82.55 | 21.14 |
| Skywork-PRM-1.5B | 84.47 | 22.45 |
| Qwen2.5-Math-PRM-7B | 85.59 | 23.48 |

result in Table 3 shows that incorporating sparsity leads to improved performance across both datasets for all PRMs, highlighting its effectiveness as a proxy signal for reward model generalization error in CATS.

## I.2 ABLATION OF HYPERPARAMETERS

**Ablation of** $\lambda_c, \lambda_m, \lambda_r$**.** The hyperparameters $\lambda_c$, $\lambda_m$, and $\lambda_r$ correspond to the coefficients of the compute cost term $C(a_t)$, the margin-based reward difference $\Delta_m(s_t, a_t)$, and the maximum predicted reward $\max_{h \in \mathcal{H}} R_\phi(q, h)$ in the reward function $r(s_t, a_t)$, respectively. We perform an ablation study on the MATH-500 dataset using Qwen2.5-Math-PRM-7B as the policy model. For each configuration, we report the average accuracy across all reward models and compute budgets. The results are summarized in Table 4. From the results in Table 4, we can observe that the best performance is achieved when all three components are present, with moderate weighting ($\lambda_c = 0.2$,

Table 4: Mean accuracy on MATH-500 under different combinations of $\lambda_c$, $\lambda_m$, and $\lambda_r$

| $\lambda_c$ | $\lambda_m$ | $\lambda_r$ | Accuracy (%) |
|---|---|---|---|
| 0.2 | 0.5 | 0.3 | **84.71** |
| 0.3 | 0.3 | 0.3 | 84.51 |
| 0 | 0.5 | 0.5 | 84.15 |
| 0.5 | 0 | 0.5 | 84.05 |
| 0.5 | 0.5 | 0 | 84.12 |
| 0 | 0 | 1 | 83.46 |
| 0 | 1 | 0 | 83.65 |
| 1 | 0 | 0 | 82.25 |

Table 5: Mean accuracy on MATH-500 under different Actor and Critic architectures. Each result is averaged over all reward models and the compute budgets.

| Actor Layers | Actor Dim | Critic Layers | Critic Dim | Accuracy (%) |
|---|---|---|---|---|
| 2 | 128 | 2 | 128 | 84.61 |
| 2 | 128 | 2 | 256 | **84.71** |
| 2 | 256 | 2 | 128 | 84.44 |
| 2 | 256 | 2 | 256 | 84.53 |
| 2 | 512 | 2 | 512 | 84.23 |
| 3 | 128 | 3 | 128 | 84.41 |
| 3 | 128 | 3 | 256 | 84.66 |
| 3 | 512 | 3 | 512 | 84.03 |

$\lambda_m = 0.5$, $\lambda_r = 0.3$). This suggests that each term in the reward function contributes to overall accuracy, and that carefully balancing these terms is essential for optimal performance. We also note that removing any single component leads to a consistent drop in accuracy. In particular, configurations that entirely exclude either $\lambda_m$ or $\lambda_r$ result in performance degradation of over 1%. This indicates that $\Delta_m(s_t, a_t)$ and $\max_{h \in \mathcal{H}} R_\phi(q, h)$ are both critical. Interestingly, the configuration with only the cost term ($\lambda_c = 1$, $\lambda_m = \lambda_r = 0$) performs the worst, highlighting that $C(a_t)$ alone is insufficient. These findings validate the design of our composite reward function and demonstrate the necessity of jointly modeling compute, ranking confidence, and reward scale.

**Ablation of Network Structure** In this section, we investigate how different architectural choices for the actor and critic networks affect the performance of CATS. We perform an ablation study on the MATH-500 dataset using Qwen2.5-Math-PRM-7B as the policy model. For each configuration, we report the average accuracy across all reward models and compute budgets. Specifically, we vary the number of layers and hidden dimensions of both networks to assess their impact on overall performance. From Table 5, we can observe that the best result is achieved when using 2-layer actor and 2-layer critic networks with hidden dimensions of 128 and 256, respectively. Increasing the hidden size beyond 256 or adding more layers does not lead to further improvement and may even result in performance degradation, possibly due to overfitting or optimization instability. These results suggest that lightweight network architectures are sufficient for effective reasoning control in CATS.

**Ablation of $\gamma$** The discount factor $\gamma$ controls the relative importance of long-term versus immediate rewards in the value estimation of the critic. To evaluate its impact, we conduct an ablation study on the MATH-500 dataset using Qwen2.5-Math-PRM-7B as the policy model. We vary $\gamma$ across a range of values and report the average accuracy across all reward models and compute budgets. The results are presented in Table 6. From the results in Table 6, we observe that the choice of the discount factor $\gamma$ has an effect on performance. Accuracy improves as $\gamma$ increases from 0.5 to 0.9, with the best result achieved at $\gamma = 0.9$. This suggests that considering future reward signals over a moderate horizon helps the critic estimate value more effectively. However, further increasing $\gamma$ beyond 0.9 leads to a slight decline in performance. These findings indicate that a moderately high discount factor strikes a good balance between immediate reward and future planning in reasoning control.

Table 6: Mean accuracy (%) on MATH-500 for different values of the discount factor $\gamma$, averaged over all reward models and compute budgets.

| $\gamma$ | 0.5 | 0.7 | 0.9 | 0.95 | 0.99 | 1.0 |
|---|---|---|---|---|---|---|
| Accuracy (%) | 84.21 | 84.33 | **84.71** | 84.63 | 84.60 | 84.56 |

## J  PSEUDO CODE FOR CATS

We provide the pseudo code for training and testing the proposed CATS algorithm. The training procedure is detailed in Algorithm 1, while the test-time inference procedure is outlined in Algorithm 2.

---

**Algorithm 1:** Actor-Critic Training in Compute-Aware Tree Search

---

**Input:** Environment $\mathcal{E}$, PRM $R_\phi$, Actor $\pi_\nu(a \mid s)$, Critic $V_\phi(s)$
**Input:** Hyperparameters: learning rate $\eta$, discount factor $\gamma$, beam size $K$, max steps $T$
**Result:** Trained actor $\pi_\nu$ and critic $V_\phi$

1 Initialize actor $\pi_\nu$ and critic $V_\phi$ with parameters from `cats_config`;
2 **for** $t = 1$ **to** $T$ **do**
3    Reset environment: $(q, a_0) \leftarrow \mathcal{E}.\text{reset}()$;
4    Initialize root node $h_0$ with state $s_0 \leftarrow \text{ExtractFeatures}(h_0)$;
5    Initialize beam $\mathcal{B}_0 \leftarrow \{h_0\}$;
6    **for** $d = 1$ **to** *max_depth* **do**
7      $\mathcal{B}_d \leftarrow \emptyset$ ;                     `// next-level beam`
8      **foreach** $h \in \mathcal{B}_{d-1}$ **do**
9        $s_t \leftarrow \text{ExtractFeatures}(h)$;
10       Sample $a_t \sim \pi_\nu(\cdot \mid s_t)$ and compute $\log \pi_\nu(a_t \mid s_t)$;
11       Expand node $h$ using action $a_t$, producing and retain children $\{h_i'\}$;
12       Compute reward $r_t \leftarrow \text{Reward}(h, \{h_i'\})$;
13       Store $(s_t, a_t, \log \pi_\nu(a_t \mid s_t), r_t)$ in $h$ for each child;
14       Add $\{h_i'\}$ to $\mathcal{B}_d$;
15      Prune $\mathcal{B}_d$ to Beam Size based on $R_\phi$;
16      **foreach** $h' \in \mathcal{B}_d$ **do**
17        $s_{t+1} \leftarrow \text{ExtractFeatures}(h')$;
18       Retrieve $(s_t, a_t, \log \pi_\nu(a_t \mid s_t), r_t)$ from parent node;
19       Compute TD-error: $\delta_t = r_t + \gamma V_\phi(s_{t+1}) - V_\phi(s_t)$;
20       Update critic: $\phi \leftarrow \phi - \eta \nabla_\phi \left(\frac{1}{2}\delta_t^2\right)$;
21       Update actor: $\theta \leftarrow \theta + \eta \nabla_\theta \left(\log \pi_\nu(a_t \mid s_t) \cdot \delta_t\right)$;

---

## K  USE OF LARGE LANGUAGE MODELS

Large language models were used solely as a general-purpose tool for grammar and language polishing. They were not involved in research ideation, methodology, analysis, or substantive writing. The authors take full responsibility for all contents of the paper.

---

**Algorithm 2:** Inference with CATS

---

**Input:** Environment $\mathcal{E}$, PRM $R_\phi$, Trained Actor $\pi_\nu$, Beam size $K$, Max steps $T$

**Result:** Set of completed reasoning paths with associated scores

**1** Initialize environment: $(q, a_0) \leftarrow \mathcal{E}.\texttt{reset}()$;

**2** Initialize root node $h_0$ with state $s_0 \leftarrow \texttt{ExtractFeatures}(h_0)$;

**3** Initialize beam $\mathcal{B}_0 \leftarrow \{(h_0, \mathcal{E})\}$;

**4** Initialize completed set $\mathcal{F} \leftarrow \emptyset$;

**5 for** $t = 1$ **to** $T$ **do**

**6**    $\mathcal{B}_{\text{next}} \leftarrow \emptyset$;

**7**    **foreach** $(h, \mathcal{E}_h) \in \mathcal{B}_{t-1}$ **do**

**8**       $s_t \leftarrow \texttt{ExtractFeatures}(h)$;

**9**       Sample action $a_t = \arg\max \pi_\nu(s_t)$ ;

**10**       Expand node $h$ using action $a_t$, generating children $\{h_i'\}$;

**11**       Retain children from $\{h_i'\}$ according to $a_t$;

**12**       **foreach** *retained* $h_i'$ **do**

**13**          Copy environment $\mathcal{E}' \leftarrow \mathcal{E}_h.\texttt{copy}()$;

**14**          Step forward: $(a, r, \texttt{done}, \_, \_) \leftarrow \mathcal{E}'.\texttt{step}(h_i'.\texttt{action})$;

**15**          **if** $\texttt{done}$ **then**

**16**             Mark $h_i'$ as terminal and add $(h_i', \mathcal{E}')$ to $\mathcal{F}$;

**17**          **else**

**18**             Further expand $h_i'$ using legal actions from $\mathcal{E}'$;

**19**             Add $(h_i', \mathcal{E}')$ to $\mathcal{B}_{\text{next}}$;

**20**    Prune $\mathcal{B}_{\text{next}}$ to top-$(K - |\mathcal{F}|)$ nodes by score;

**21**    $\mathcal{B}_t \leftarrow \mathcal{B}_{\text{next}}$;

**22**    **if** $|\mathcal{F}| = K$ **then**

**23**       **break**

**24** Return completed set $\mathcal{F}$ as final trajectories;

---

