# OpenReview forum: "Reward Model Generalization for Compute-Aware Test-Time Reasoning"
_ICLR.cc/2026/Conference — ICLR 2026 Conference Withdrawn Submission_

### Official Review · Reviewer_Rikj · 2025-10-25

**Soundness:** 2
**Presentation:** 2
**Contribution:** 2
**Rating:** 2
**Confidence:** 3

**Summary:**

This paper presents Compute-Aware Tree Search (CATS), a test-time algorithm to extract answers assuming the access to PRMs.

### Theoretical Framework
The work introduces a theoretical framework to analyze the core challenge of Test-Time Compute Optimality (TCO).

1.  PRM Generalization Analysis: The authors leverage PAC-Bayes theory to derive generalization bounds for the PRM, defining the generalization error ($\epsilon_{\text{gen}}(\phi)$) as the deviation between the population risk and the empirical risk.
2.  Accuracy Bounds: A central finding (Theorem 4.5) establishes an explicit lower bound on answer accuracy by quantifying the risk of mis-ranking candidate reasoning paths due to PRM prediction errors. This lower bound is jointly determined by three components:
    - The probability of sampling at least one correct path ($p_{N, \tau}(q)$).
    - The reward gap ($\gamma(q)$) between the path selected by the true best path and the next best discarded path.
    - The upper bound of the generalization error ($\epsilon$).
3.  Compute Budget Insight: The analysis shows that a higher generalization error of the PRM necessitates sampling more reasoning paths to guarantee a desired accuracy level ($\alpha$), demonstrating the critical impact of PRM generalization on the required compute budget.

### Methodology
Motivated by the theory, which highlights that the generalization error ($\epsilon$) is unobservable at test time, the authors propose CATS, a dynamic reasoning control strategy.

- CATS addresses the unobservability of $\epsilon$ by using the structural sparsity ($\|\phi\|_0/d$) of the PRM parameters as a practical and observable proxy for generalization capacity.
- The reward function $r(s_t, a_t)$ is composite, designed to optimize for TCO by jointly modeling: (1) compute cost $C(a_t)$, (2) the margin-based reward difference between retained and discarded paths $\Delta m(s_t, a_t)$, and (3) the maximum predicted reward $\max_h R_{\phi}(q, h)$.

### Experimental Validation

Extensive experiments were conducted on two challenging mathematical reasoning benchmarks, MATH-500 and AIME24, across diverse sets of frozen policy models (e.g., Llama 3.1, Qwen 2.5) and various PRMs (e.g., Math-Shepherd-PRM-7B, Skywork-PRM-1.5B).

**Strengths:**

- there are many theoretical results
- graphs are clear and get the point across

**Weaknesses:**

1. the performance gain over beam search seem quite marginal
2. the lower bound on the probability that the policy model generates a correct answer seems quite trivial, I don't think this can qualify as a contribution
3. if you're doing this strategy, then at each action step, you would need to stop and calculate the next best action to choose and subsequently send in a new query, correct? In practice, is this really feasible? Since sending a new query may lead to scheduling, KV cache, and synchronization overheads.

**Questions:**

1. How does your method compare against test-time scaling methods beyond beam search?
2. How feasible is the deployment in practice? Under what scenarios would this be used? See weakness 3.
3. Could you show these results on newer models such as Qwen3, that have longer reasoning budgets? Is it still computationally feasible to run the algorithm on such models?

---

### Official Review · Reviewer_rp7R · 2025-10-30

**Soundness:** 2
**Presentation:** 3
**Contribution:** 2
**Rating:** 4
**Confidence:** 4

**Summary:**

This paper investigates how to improve reasoning performance for large language models (LLMs) under a limited test-time compute budget. The authors present a PAC-Bayes–inspired theoretical framework that relates the generalization behavior of process reward models (PRMs) to answer accuracy, reward gaps, and sampling configurations. Building upon these insights, the paper proposes Compute-Aware Tree Search (CATS), an inference-time reasoning strategy formulated through an Advantage Actor–Critic (A2C) paradigm. CATS dynamically adjusts sampling parameters (e.g., temperature, beam width) during rollout generation. Experiments on MATH500 and AIME24 suggest consistent improvements across several policy LLMs and PRMs under constrained rollout counts.

**Strengths:**

**Originality**
The paper offers a theoretically grounded perspective on inference-time reasoning that connects reward gap geometry with sampling decisions. Positioning adaptive rollout allocation as a policy-optimization problem is an interesting and under-explored angle. The PAC-Bayes analysis provides conceptual clarity on when larger rollout budgets are beneficial and yields intuitive design implications.

**Quality**
The method is clearly described, and experiments across multiple datasets, policy models, and PRMs demonstrate that CATS tends to outperform standard choices such as Best-of-N and static sampling heuristics. The qualitative analysis of reward gaps and sparsity is useful for interpreting model behavior.

**Weaknesses:**

**1. Strength of empirical baselines.**
While the paper includes common external TTS strategies, recent strong baselines such as DVTS [1] and REBASE [2] have been shown to be competitive under similar compute constraints. Without comparisons to these methods, it is difficult to judge the relative advantage of CATS. The gains reported here are modest in some configurations, especially when the underlying policy model is relatively strong, raising questions about the robustness of improvements.


**2. Compute fairness and budgeting.**
Although the core evaluation constrains rollout counts, the proposed approach requires additional data collection and training of actor–critic components, incurring non-trivial compute cost. This overhead is not accounted for when comparing against plug-and-play methods. A more rigorous compute accounting (e.g., via FLOPs) would contextualize practical applicability.

**3. Transferability and policy mismatch.**
The paper does not clearly illustrate how well the learned adaptation policy generalizes across different underlying LLMs. In realistic deployment scenarios, downstream users may substitute or upgrade policy models. It is unclear whether CATS needs retraining in such cases and how performance degrades under mismatch.

**4. Theory–practice gap.**
The training objective does not directly optimize the derived lower bound but instead uses surrogate linear combinations. This gap is understandable but should be stated more explicitly, since the theoretical framing suggests a more principled objective than what is actually implemented.

**5. Missing discussion vs. closely related recent work.**
Recent work such as [3] studies optimal allocation of rollouts across reasoning paths with a mathematical treatment of compute-aware scheduling. The current paper appears related in motivation but does not position itself relative to these findings. A discussion is needed to clarify conceptual distinctions.

[1] Scaling test-time compute with open models.

[2] Inference scaling laws: An empirical analysis of compute-optimal inference for llm problem-solving. ICLR 2025

[3] Every Rollout Counts: Optimal Resource Allocation for Efficient Test-Time Scaling. NeurIPS 2025

**Questions:**

1. **Baseline selection:**
   Can the authors add comparisons against recent external TTS methods such as DVTS [1] and REBASE [2]?

3. **Compute overhead:**
   What is the total training and data collection cost of learning the actor–critic components, expressed in FLOPs or GPU hours? How does this compare to inference-only strategies?

4. **Theory–practice connection:**
   Could the authors elaborate on why Equation (6) cannot be optimized directly and how the chosen surrogate reflects the theorem’s assumptions?

5. **Relation to [3]**
   How does CATS differ conceptually from the optimal allocation perspective proposed in **Every Rollout Counts** [3]? Are the two approaches compatible or contradictory in their assumptions?

[1] Scaling test-time compute with open models.

[2] Inference scaling laws: An empirical analysis of compute-optimal inference for llm problem-solving. ICLR 2025

[3] Every Rollout Counts: Optimal Resource Allocation for Efficient Test-Time Scaling. NeurIPS 2025

---

### Official Review · Reviewer_WBAR · 2025-10-30

**Soundness:** 2
**Presentation:** 2
**Contribution:** 2
**Rating:** 6
**Confidence:** 2

**Summary:**

The paper studies how a PRM’s generalization impacts accuracy and compute in external TTS, derives bounds (via PAC-Bayes) linking accuracy to reward gap and PRM error, and proposes CATS—an RL controller that dynamically allocates inference compute to enlarge effective reward margin and improve accuracy under fixed budget.

**Strengths:**

- The paper provides a clear theoretical analysis of how the generalization error of the PRM affects compute efficiency and reasoning performance.
- Building on the above theoretical findings, the paper designs CATS, and demonstrates effective downstream performance consistent with the theory.

**Weaknesses:**

- The baseline comparison is not sufficiently comprehensive — the paper does not include comparisons with several recent and relevant test-time scaling methods such as DVTS [1], REBASE [2], and DORA [3], which also study compute allocation and inference efficiency. Without these, it is difficult to gauge how much of the observed improvement comes from the proposed controller versus general dynamic inference strategies.

- The theoretical results are based on a set of reasonable assumptions, which may not fully hold in practical scenarios


[1] Scaling Test-Time Compute with Open Models

[2] Inference Scaling Laws: An Empirical Analysis of Compute-Optimal Inference for LLM Problem-Solving. ICLR 2025

[3] Every Rollout Counts: Optimal Resource Allocation for Efficient Test-Time Scaling. NeurIPS 2025

**Questions:**

See weakness.

---

### Note · Authors · 2025-11-19

I have read and agree with the venue's withdrawal policy on behalf of myself and my co-authors.